# mAb14, a Monoclonal Antibody against Cell Surface PCNA: A Potential Tool for Sezary Syndrome Diagnosis and Targeted Immunotherapy

**DOI:** 10.3390/cancers15174421

**Published:** 2023-09-04

**Authors:** Jamal Knaneh, Emmilia Hodak, Shlomit Fedida-Metula, Avishay Edri, Rachel Eren, Yael Yoffe, Iris Amitay-Laish, Hadas Prag Naveh, Ido Lubin, Angel Porgador, Lilach Moyal

**Affiliations:** 1Laboratory for Molecular Dermatology, Felsenstein Medical Research Center, Tel Aviv 6997801, Israel; jamal.knaneh@gmail.com (J.K.); hodake@post.tau.ac.il (E.H.); 2Sackler Faculty of Medicine, Tel Aviv University, Tel Aviv 6997801, Israel; amitaylaishiris@gmail.com (I.A.-L.); hadas.prag.naveh@gmail.com (H.P.N.); 3Davidoff Cancer Center, Rabin Medical Center, Petach Tikva 4941492, Israel; 4PiNK Biopharma Ltd., Ness Ziona 7403648, Israel; fedida.metula@gmail.com (S.F.-M.); yael.yoffe@gmail.com (Y.Y.); 5The Shraga Segal Department of Microbiology, Immunology and Genetics, Faculty of Health Sciences, Ben-Gurion University of the Negev, Beer Sheva 8410101, Israel; avishayedri@gmail.com (A.E.); angel@bgu.ac.il (A.P.); 6Division of Dermatology, Rabin Medical Center, Petach Tikva 4941492, Israel; 7Core Facility, Felsenstein Medical Research Center, Rabin Medical Center, Sackler Faculty of Medicine, Tel Aviv University, Tel Aviv 6997801, Israel; idolu@tauex.tau.ac.il; 8National Institute for Biotechnology in the Negev, Ben Gurion University of the Negev, Beer Sheva 8410101, Israel

**Keywords:** cutaneous T cell lymphoma (CTCL), *Mycosis fungoides* (MF), *Sézary syndrome* (SS), proliferating cell nuclear antigen (PCNA), monoclonal antibody 14 (mAb14), natural killer p44 isoform-1 (NKp44-1)

## Abstract

**Simple Summary:**

We need better ways to detect and treat skin lymphoma, a type of skin cancer. Current treatments that enhance the body’s cancer-fighting abilities are not very effective, and the markers used for diagnosis are not specific. We’ve previously found a new marker called PCNA on the surface of various cancer cells. PCNA seems to stop immune cells from killing cancer cells. Investigating PCNA on skin lymphoma cells could help with diagnosis and improving treatment. To do this, we’re using a special homemade antibody called mAb14 to study PCNA on the outside of skin lymphoma cells. We’re also checking if blocking PCNA with mAb14 affects the immune cells’ ability to kill these cancer cells. Our research suggests that mAb14 could be a useful tool for detecting skin lymphoma cells in the blood of patients with advanced disease. It may also help boost the immune system’s fight against skin lymphoma through immunotherapy.

**Abstract:**

*Mycosis fungoides* (MF) and *Sézary syndrome* (SS) are the most common types of primary cutaneous T-cell lymphoma (CTCL). Proliferating cell nuclear antigen (PCNA) is expressed on the cell surface of cancer cells (csPCNA), but not on normal cells. It functions as an immune checkpoint ligand by interacting with natural killer (NK) cells through the NK inhibitory receptor NKp44, leading to the inhibition of NK cytotoxicity. A monoclonal antibody (mAb14) was established to detect csPCNA on cancer cells and block their interaction with NKp44. In this study, three CTCL cell lines and peripheral blood mononuclear cells (PBMCs) from patients with SS and healthy donors were analyzed for csPCNA using mAb14, compared to monoclonal antibody PC10, against nuclear PCNA (nPCNA). The following assays were used: immunostaining, imaging flow cytometry, flow cytometry, cell sorting, cell cycle analysis, ELISA, and the NK-cell cytotoxic assay. mAb14 successfully detected PCNA on the membrane and in the cytoplasm of viable CTCL cell lines associated with the G2/M phase. In the Sézary PBMCs, csPCNA was expressed on lymphoma cells that had an atypical morphology and not on normal cells. Furthermore, it was not expressed on PBMCs from healthy donors. In the co-culture of peripheral blood NK (pNK) cells with CTCL lines, mAb14 increased the secretion of IFN-γ, indicating the reactivation of pNK activity. However, mAb14 did not enhance the cytotoxic activity of pNK cells against CTCL cell lines. The unique expression of csPCNA detected by mAb14 suggests that csPCNA and mAb14 may serve as a potential biomarker and tool, respectively, for detecting malignant cells in SS and possibly other CTCL variants.

## 1. Introduction

Primary CTCL is a type of non-Hodgkin lymphoma caused by the clonal expansion of malignant T lymphocytes in the skin. The most common forms of CTCL are MF and SS. Each of these arises from distinct T-cell subsets: MF from skin-resident effector memory T-cells and SS from central memory T-cells [1]. The diagnosis of CTCL is difficult in the early stages because of its multiple clinical presentations [2,3,4] and the lack of definitive diagnostic criteria [2,5].

Numerous molecular biomarkers are associated with CTCL, providing significant insights into disease origin, accurate diagnosis, prognosis evaluation, and optimal treatment selection. The following is a concise summary of promising potential biomarkers for CTCL: (1) The high expression of KIR3DL2/CD158k, primarily found on NK cells and a minor subset of CD4 and CD8 T cells, is observed in all CTCL variants. This expression is concurrent with disease staging and corresponds to reduced overall survival (OS) rates [6,7,8,9,10,11,12,13,14]. (2) CD30 expression, categorized as a cytokine receptor within the tumor necrosis factor receptor superfamily, is present in a specific subset of CTCL known as CD30-positive primary CTCL (PC CD30+ T-LPD) [15,16,17,18,19,20,21,22]. (3) A significant number of SS patients exhibit a combination of PLS3, Twist, KIR3DL2, and NKp46 expression [23]. (4) TOX is overexpressed in both the lesional skin and peripheral blood of SS patients, carrying prognostic implications [24,25,26,27,28]. (5) The high expression of EPHA4 is noted in SS patients [29]. (6) The increased expression of CD164 is observed in CD4+CD26− T cells of SS patients [30]. (7) Increased FCRL3 expression is found in SS patients with a high tumor burden, which correlates with disease progression [30]. (8) Soluble IL-2 receptor serves as a potential marker for the activity, severity, and prognosis of MF [31]. (9) Syndecan-4 (SD-4) is overexpressed in CTCL cell lines and malignant CD4+ T cells of SS [32,33]. (10) CADM1 expression is observed in T-cell lines derived from various CTCL subtypes, including MF and SS [34,35,36,37]. (11) The high expression of CXCL13 is detected in the skin lesions, lymph nodes, and plasma of patients with SS, which is also common in MF [38,39,40,41]. However, none of these biomarkers are currently in clinical use. For assessing the involvement of blood in SS, the ISCL/USCLC/EORTC consortium recommends using CD4+CD7- and/or CD4+CD26- lymphocytes [42].

Proliferating cell nuclear antigen (PCNA) is a well-documented protein that is frequently overexpressed in the nuclei of cancer cells, and this overexpression has been linked to increased cancer virulence. PCNA plays a significant role in promoting the cancerous phenotype by actively participating in crucial processes that are essential for the survival of tumor cells. These processes encompass DNA replication, the repair of DNA damage, the preservation of chromatin architecture, accurate chromosome segregation, and the progression of the cell cycle [43,44].

Natural killer (NK) cells play a crucial role in the primary innate immune defense against cancer. Activated NK cells express NKp44, a natural cytotoxicity receptor [45,46,47] that exerts cytotoxic activity against both tumor and virus-infected cells [48,49]. Studies have shown decreased NK cell activity in CTCL, accompanied by the down-regulation of NK-activating receptors, NKG2C, and NKG2D [50,51,52].

Prof. Porgardor’s research team previously revealed that the presence of NKp44 on NK cells leads to the recruitment of the fraction of PCNA localized on the plasma membrane of cancer cells into the NK immunological synapse [43,49,53]. The subsequent interaction of the cell surface PCNA (csPCNA) with NKp44 initiates a potent signaling cascade that is predominantly mediated by immune receptor tyrosine-based inhibitory motifs (ITIMs), which inhibit the killing activity of NK cells against cancer cells [43,53,54,55,56,57,58,59]. To inhibit the interaction between NKp44 and csPCNA, Kundu et al. developed a mouse monoclonal antibody against csPCNA, named mAb14 [53]. The antibody successfully detected csPCNA on the surface of various cancer cell lines, including B-cell lymphoma and chronic myelogenous leukemia, while showing no reactivity with matched healthy cells [53]. Co-culturing the NK cell line with several cancer cell lines, along with the addition of mAb14, enhanced the NK cell anti-cancer activity in terms of interferon gamma (IFN-γ) secretion and cancer cell cytotoxicity. Furthermore, in mice carrying patient-derived xenografts of head and neck squamous carcinoma, systemic treatment with mAb14 and human NK cells resulted in inhibited tumor growth [53].

PCNA migration towards the membrane of tumor cells might initiate the physical interaction between the NK cells and target cells, or the lytic state of other tumor cells attacked by NK cells. It is also possible that PCNA is carried to or anchored at the membrane by an associated protein such as Annexin A2 (ANXA2) [57,60]. The downregulation of Annexin A2 has been reported in several cancers (esophageal squamous cell carcinoma, oral squamous cell carcinoma, and prostate cancer) [61,62], and might have an effect on PCNA migration to the membrane.

A recent study also demonstrated the expression of PCNA on the cell surface of triple-negative breast cancer cells (TNBCs), which interact with NK cell receptors through NKp44, leading to the inhibition of the NK cell cytolysis function against cancer cells. Disrupting the PCNA–NKp44 interaction by blocking PCNA on TNBC cells using specific antibody effectively enhanced the lysis of TNBC cells by the primary NK cells [63].

The present study aims to investigate the expression of csPCNA in CTCL cell lines and SS blood by the monoclonal antibody mAb14, and to explore the potential of mAb14 as a diagnostic biomarker for CTCL and a target for NK-mediated immunotherapy.

## 2. Materials and Methods

Three CTCL cell lines (Hut78, Myla, HH) and peripheral blood mononuclear cells (PBMCs) from patients with SS and healthy donors were analyzed for csPCNA using mAb14, compared to monoclonal antibody PC10, against nuclear PCNA (nPCNA). Hut78 cells are lymphoma cells derived from the PBMCs of SS patients, MyLa are lymphoma cells derived from the skin biopsy of MF patients, and HH cells are lymphoma cells derived from the PBMCs of patients with aggressive CTCL [64].

### 2.1. Isolation of PBMCs from Patients with SS and Healthy Controls

PBMCs were collected under local Helsinki approval for the bioethical permission of Rabin Medical Center: Ref. 7175 for the blood of 9 patients with SS, and Ref. 6515 for the leftover blood of healthy donors. Patient’s clinical data are presented in Appendix A. PBMCs were isolated using ficol, as was previously described [65].

### 2.2. Isolation and Culture of Primary Human Natural Killer (NK) Cells

Primary human NK cells were isolated from the peripheral blood of healthy donors using the RosetteSep Human NK Cell Enrichment Cocktail 15,065 (STEMCELL Technologies, Vancouver, BC, Canada), as previously described [64].

### 2.3. Antibodies

The antibodies used were as follows: mAb14 (PINK Biopharma, Ness Ziona, Israel) anti-csPCNA at 400 ng/10^5^ cells (unless otherwise noted) combined with streptavidin-APC (Miltenyi Biotec, Woking, UK) at 1–2 µg/mL and biotinylated anti-mouse IgG1ĸ (1:50) (BioLegend, San Diego, CA, USA) for isotype control; PC10 (BioLegend, San Diego, CA, USA) anti-nuclear PCNA (nPCNA) at 1 µg/mL (unless otherwise noted) combined with Alexa Fluor 488 conjugated goat anti-mouse IgG at 1 µg/mL (Jackson ImmunoResearch, West Grove, PA, USA) and its matched isotype control (IgG2a) (BioLegend, San Diego, CA, USA).

### 2.4. Immunofluorescence Staining of csPCNA and nPCNA

Cells were stained with mAb14 and combined with Alexa Fluor 568 donkey anti-mouse at 1 µg/mL (Life Technologies, Carlsbad, CA, USA), as previously described [43]. Findings were analyzed with the Axio Imager Z2 Microscope (Zeiss, Jena, Germany) (×40 magnification).

### 2.5. Imaging Flow Cytometry for csPCNA and nPCNA

Cells were stained with mAb14 and PC10 as previously described [53], and analyzed with the high-resolution multispectral ImageStream (Acq) imaging flow cytometer (Amnis), which produces multiple images of each cell directly in flow (×40 magnification).

### 2.6. Flow Cytometry for csPCNA and nPCNA

Cells were stained with mAb14 and PC10 as previously described [43], and analyzed with the Gallios Flow Cytometer (Beckman Coulter, Brea, CA, USA) on live cells (DAPI^+^) using Kaluza 1.3 software.

### 2.7. Apoptosis Assay

Cells were stained with recombinant mAb14 at 5 µg/mL, or with the isotype control and then with Annexin V FITC (eBioscience, San Diego, CA, USA) and propidium iodide (PI) (eBioscience, San Diego, CA, USA). Florescence-activated cell sorting (FACS) analysis was performed with the Gallios flow cytometer using Kaluza 1.3 software. The mAb14+ cells were analyzed for apoptosis based on the Annexin and PI double staining.

### 2.8. Cell Cycle Analysis of Cells Expressing csPCNA (mAb14+)

Cells were stained with mAb14 and DAPI (Sigma, St. Louis, MO, USA). The DAPI-stained mAb14+ and mAb14− cell populations were sorted with the BD FACS Aria III flow cytometer (BD Biosciences, Franklin Lakes, NJ, USA) and collected in 2% fetal bovine serum (FBS). The sorted cells and a fraction of unsorted cells were washed and analyzed for cell cycle distribution with the Gallios flow cytometer, as previously described [65]. Data were analyzed according to an algorithm using Kaluza 1.3 software.

### 2.9. G2/M Cell Cycle Arrest by Nocodazole

Cells were treated with Nocodazole (Sigma, St. Louis, MO, USA), 80 ng/mL for Hut78 cells and 100 ng/mL for MyLa and HH cells, and incubated for 24 h or 48 h, respectively. One sample of the treated cells was stained with mAb14 and analyzed using fluorescence microscopy, and another sample was stained with PI for flow cytometry analysis to detect cell cycle arrest.

### 2.10. Flow Cytometry for csPCNA Detection in Human Blood Samples

PBMCs from patients with SS and healthy controls were washed with FACS buffer (PBS–Mg–Ca, 2% FBS) and suspended in FACS buffer with Fc receptor (FCR) blocker and TANDEM enhancer (Miltenyi Biotec, Woking, UK) on ice for 10 min. Cells were stained with either recombinant mAb14 at 5 µg/mL or biotinylated mAb14 at 5 µg/mL, or matched isotype-control mouse IgG1ĸ at 5 µg/mL for 2 h. Cells were washed and stained with either APC AffiniPure F (ab’) 2 fragment goat anti-mouse IgG or streptavidin-APC for 45 min on ice. Cells were then washed and suspended in FACS buffer with CD4-FITC (Miltenyi Biotec, Woking, UK) and CD26-PE (Miltenyi Biotec, Woking, UK) for 30 min on ice, washed in FACS buffer, and suspended in FACS reader buffer. DAPI viability staining solution was used to distinguish between live and dead cells. ∆ positive staining for mAb14 = (%mAb14 + cells) − (%isotype control + cells).

### 2.11. Hematoxylin and Eosin (H&E) Staining of CD4+CD26- mAb14+ Peripheral Lymphocytes of Patients with SS

PBMCs from patients with SS were stained for csPCNA. Stained cells were sorted using FACS sorter (BD FACS Aria III) into two subpopulations: DAPI-CD4+CD26-mAb14+ and DAPI-CD4+CD26-mAb14−. Cells were transferred to Superfrost Plus Slides using cytospin and fixed with 4% paraformaldehyde for 10 min, followed by washing with phosphate-buffered saline (PBS). Slides were stained with H&E (American Mastertech Scientific, Lodi, CA, USA) for 30 s, washed three times with deuterium-depleted water (DDW), air dried, and covered with a cover glass. Images were taken with a light microscope (Olympus, Tokyo, Japan) (magnitude ×10).

### 2.12. NK Stimulation and IFNγ Assay

Target CTCL cells of 1.5 × 10^5^ were incubated with biotinylated mAb14 at 10 µg/mL, biotin Isotype mouse IgG1ĸ (BioLegends, San Diego, CA, USA), or no antibody for 1 h at room temperature. A total of 5 × 10^4^ cells of primary NK or NK92-MI (effector) cells were then co-cultured with these target cells (effector/target ratio, 1:3) in 96-well U-bottom plates for 20 h in an incubator. The concentration of secreted IFNγ was analyzed in cell supernatant using an ELISA MAX kit (BioLegend) and read at 650 nm with a microplate reader (BioTek, Winooski, VT, USA). Since the IFN-γ level in the co-culture of pNK cells with the CTCL cell line is attributed by to the two kind of cells, with no option to discriminate between them, we compared it to the sum of IFN-γ secretion according to each cell alone.

### 2.13. NK-Cell Cytotoxic Activity

Cells were labeled with carboxyfluorescein diacetate succinimidyl ester (CFSE; Thermo Scientific, Waltham, MA, USA), washed twice with PBS, and suspended in PBS. CFSE was added to a final concentration of 10 µM, and cells were incubated for 10 min at room temperature in the dark. Labeling was terminated by adding 4–5 volumes of cold complete medium containing ≥ 10% FBS and incubating on ice for 5 min. Cells were washed three times with medium and cultured in a 96-well plate for 1 h on ice with either 10 µg/mL of biotinylated mAb14, biotin Isotype mouse IgG1ĸ, or no antibody. Target cells were co-incubated with NK92-MI effector cells at 1.5 × 10^6^ (effector/target ratio of 1:1 or 3:1) in the presence of mAb14, its matched isotype, or no antibody for 6 h in a 96-well plate. The co-culture of target and effector cells was washed with PBS and incubated with 7-AAD staining solution (Biotest, Dreieich, Germany). The necrotic target cell quantification of CFSE^+^ (CTCL cells) and 7AAD^+^ (dead cells) was based on the flow cytometer analysis using Kaluza 1.3 software.

### 2.14. Statistical Analysis

The significance of the differential effects among the comparative groups was determined using a two-tailed, paired and unpaired *t*-test or Mann–Whitney test, one-way ANOVA in Excel for Windows 2007 (Microsoft), and GraphPad prism 8.
(1)$$\mathrm{Fold change}=\frac{absolute\;IFNg\;secretion\left[\frac{\rho g}{mL}\right]seperately}{absolute\;IFNg\;secretion\left[\frac{\rho g}{mL}\right]co\text{-}cultured}$$

The statistical significance was calculated as the percent of cells positive for mAb14 and isotype IgG1k based on the paired *t*-test (* *p* < 0.05; ** *p* < 0.01) and error bars for mean ± SD.

## 3. Results

### 3.1. mAb14 Detects PCNA on the Membrane and in the Cytoplasm of CTCL Cell Lines

CTCL cell lines (Hut78, Myla, HH) were stained with mAb14 and PC10 antibodies. The cytoplasmic and membrane mAb14 staining of PCNA was demonstrated in all three lines (Figure 1A–C, upper panel). The PC10 staining of PCNA was exhibited in the nucleus (Figure 1A–C, middle panel). There was no staining in the control cells (Figure 1A–C, lower panel).

To determine whether the fixation procedure interfered with the PC10’s ability to recognize csPCNA, we conducted PCNA staining in fixed and un-fixed cells using imaging flow cytometry. In fixed CTCL cell lines stained with PC10 and 7-AAD, we observed the nuclear staining of PCNA (Figure 1D). The positive 7-AAD staining in the nucleus indicated the occurrence of membrane perforation during fixation. In contrast, unfixed cells showed negative staining for 7-AAD and PC-10 (Figure 1E), suggesting that PC10 did not detect csPCNA. However, mAb14 successfully detected csPCNA in permeabilized cells (DAPI^+^) (Figure 1F) and in unpermeabilized cells (DAPI^−^) (Figure 1G). Evidently, the detection of csPCNA by mAb14 and of nPCNA by PC10 did not rely on the integrity of the cell membrane during the staining procedure.

### 3.2. csPCNA Is Expressed in a Subpopulation of Viable CTCL Cell Lines whereas nPCNA Is Expressed in Most of the Cells

CTCL cell lines (Hut78, Myla, HH) were stained with DAPI and mAb14 without fixation and permeabilization, and then analyzed using flow cytometry. The subpopulations of mAb14+ cells vs. the isotype control were as follows: Hut78, 28.74% (*p* = 0.0078); MyLa, 3.96% (*p* = 0.0286), and HH, 1.88% (*p* = 0.0287) (Figure 2A,B). Figure 2A shows the staining plots and representative histograms of the cell counts, and Figure 2B shows the column curves for the average percent of stained cells, demonstrating higher positive staining in the SS cell line than in the other two CTCL cell lines. Flow cytometry analysis using PC10 revealed negative staining on unfixed unpermeabilized cells (Appendix A) and positive staining in >50% of fixed permeabilized cells (Appendix A). To determine whether mAb14^+^ cells are viable and nonapoptotic, the apoptotic cells among the cells positive for mAb14 (Figure 2C) or for the isotype control (Figure 2D) were analyzed. The mAb14^+^ cells, indicated by black dots in the figure (Figure 2E), mostly appeared in the Annexin V-/PI quadrant of viable cells (Figure 2F). Most (80–95%) of the CTCL cells expressing csPCNA were viable and non-apoptotic. There was a significant difference between the mAb14^+^ viable and nonviable cells in each cell line: *p* = 3 × 10^−6^ Hut78; *p* = 9 × 10^−6^ MyLa; and *p* = 1.9 × 10^−5^ HH (Figure 2F).

### 3.3. csPCNA Is Mostly Expressed in CTCL Cells at G2/M Phase

To determine whether there is a correlation between the cell cycle phase and expression of csPCNA in CTCL, CTCL cell lines were stained with DAPI and mAb14. Viable (DAPI^−^) mAb14^+^ cells and viable (DAPI^−^) mAb14^−^ were sorted using FACS. Unsorted cells and the two sorted cell populations were then fixed and stained with PI for cell cycle analysis using flow cytometry (Figure 3A). The average proportions of cells in the G1 and G2/M phase among the viable mAb14^+^ cells compared to the viable mAb14^−^ cells were calculated (Figure 3B). The viable mAb14+ group had a significantly lower percentage of cells in the G1 phase: Hut78, 33.19% vs. 56.45% (*p* = 0.0135); MyLa, 38.70% vs. 66.24% (*p* = 0.0680), and HH, 22.72% vs. 56.63% (*p* = 0.0255) (Figure 3B), and a higher percentage of cells in the G2/M phase: Hut78, 52.44% vs. 29.34% (*p* = 0.0138); MyLa, 50.77% vs. 24.73% (*p* = 0.081); and HH, 67.19% vs. 31.65% (*p* = 0.0208) (Figure 3B). The results suggested that the expression of csPCNA in CTCL cell lines increased in the G2/M cell cycle phase.

To confirm this assumption, cells were treated with Nocodazole for G2/M arrest. The untreated and Nocodazole-treated cells were split for two analyses: cell cycle analysis to confirm G2/M arrest (Appendix A), and flow cytometer analysis for mAb14 and isotype control staining (Figure 3C–H). We detected an increase in mAb14 staining in the G2/M-arrested cells (Nocodazole treated) compared to the untreated cells, as follows: Myla, ∆ = 21.727% (*p* = 0.0393) (Figure 3G) and HH, ∆ = 17.047% (*p* = 0.0127) (Figure 3H). In the SS cell line (Hut78), with high basal mAb14 staining, Nocodazole induced a non-significant increase of mAb14 staining (∆ = 4.137%, *p* = 0.6920) (Figure 3F).

### 3.4. Viable CD4^+^CD26^−^mAb14^+^ Peripheral Lymphocytes of Patients with Sezary Exhibit Atypical Morphology

PBMCs from patients with SS (n = 3) were stained with CD4-FITC, CD26-PE, DAPI, and mAb14-biotin-APC. The DAPI^−^CD4^+^CD26^−^ (defined as the SS-enriched cell population [15,42]) mAb14^+^ cells and the DAPI^−^CD4^+^CD26^−^ mAb14^−^ cells were collected using FACS sorter. The two sorted cell populations were attached to slides using cytospin and stained with H&E (Figure 4A). The morphology of the DAPI^−^CD4^+^CD26^−^ mAb14^+^ cells was mostly convoluted and atypical, with moderately to highly grooved (i.e., cerebriform) nuclei and a high nuclear to cytoplasmic ratio, as is typical for the lymphoma cells of SS (indicated by arrow heads) [66]. Meanwhile, the DAPI^−^CD4^+^CD26^−^ mAb14^−^ subpopulation was mostly a mixture of round-shaped non-cerebriformic normal lymphocytes (indicated by arrow heads). The ∆%mAb14+ of DAPI^−^CD4^+^CD26^−^ cells in SS patients compared to healthy individuals is presented in a scatter plot (Figure 4B).

### 3.5. csPCNA Is Expressed in PBMCs from Patients with Sezary but Not PBMCs from Healthy Donors

The expression of csPCNA in SS PBMCs (n = 3) and the healthy control (n = 3) was analyzed using immunofluorescence staining with biotinylated mAb14. The cytoplasmic and membrane staining of PCNA is shown (Figure 5A–C, upper panel). The PC10 antibody exhibited nuclear staining (Figure 5A–C, middle panel). There was no staining with mAb14 in the PBMCs of the healthy controls (Figure 5D–F, upper panel), and the weak staining of nPCNA, as expected in normal cells (Figure 5D–F, middle panel).

PBMCs from five SS patients were stained with CD4-FITC, CD26-PE, DAPI, and mAb14, followed by flow cytometry analysis. Mouse IgG1ĸ was used as a control. Patient PBMCs contained 30–70% viable CD4^+^CD26^−^ peripheral lymphocytes (Figure 5G, left panel). Among the lympho-gated viable DAPI^−^CD4^+^CD26^−^ cells [15,42] (considered mostly malignant), 2.76–13.55% were positive for mAb14 (Figure 5G, middle panel) compared to less than 1% of the isotype control cells (Figure 5G, right panel). The same experiment performed on the blood samples of five healthy donors yielded a low proportion (<10%) of DAPI^−^CD4^+^CD26^−^ cells (Figure 5H, left panel) with the negative staining of mAb14 (Figure 5H, middle panel); this is compared to the isotype controls (less than 1%) (Figure 5H, right panel).

### 3.6. mAb14 Activates IFN-γ Secretion from Effector Primary NK Cells Co-Cultured with CTCL Cell Lines

Since csPCNA interacted with NKp44 on NK cells, and mAb14 interfered with that interaction, the next step was to determine whether CTCL cell lines inhibit the secretion of IFN-γ by pNK (effector) cells and whether the mAb14 antibody reduces this inhibition. First, pNK cells from healthy donors were analyzed for the presence of NKp44 using flow cytometry. The CD3-Vio-blue, CD56-PE, and NKp44-APC values were compared to the isotype controls (Figure 6A). Thereafter, the level of IFN-γ secreted by CTCL cell lines and pNK was determined using ELISA, showing that pNK cells express a high level of IFN-γ, while all three CTCL cell lines express a lower to negligible level compared to pNK: Hut78 (*p* = 0.0014), Myla (*p* = 0.0014), and HH (*p* = 0.0014) (Figure 6B).

CTCL (target) cells were incubated with pNK (effector) cells for 20 h at a ratio of 3:1, and then analyzed for the secretion of IFN-γ using ELISA. The level of secreted IFN-γ was reduced when cells were co-cultured relative to the sum of the absolute IFN-γ secreted by each cell alone, as follows: Hut78+pNK, decreased by 1.92-fold (*p* = 0.0650); MyLa + pNK, decreased by 1.27-fold (*p* = 0.0255); and HH + pNK, decreased by 1.76-fold (*p* = 0.0369). The column curves in the figure show the absolute IFN-γ secretion occurring in three independent experiments using both separate and combined cultures (Figure 6C). Finally, to test whether mAb14 eliminates the CTCL-mediated reduction in the secretion of IFN-γ by pNK cells, CTCL (target) cells at 1.5 × 10^5^ were incubated with 10 µg/mL of biotinylated mAb14, with biotin Isotype mouse IgG1ĸ as a control for 1 h. The pNK effector cells were then co-cultured with the CTCL target cells at a ratio of 1:3 for 20 h. IFN-γ secretion in the co-cultured cells was determined using ELISA. A significant increase in IFN-γ secretion was observed when pNK cells were co-cultured with mAb14 compared to the isotype control, as follows: Hut78 + pNK, 1.37-fold (*p* = 0.0339); MyLa+ pNK, 1.34-fold (*p* = 0.0208); HH+ pNK, 2.06-fold (*p* = 0.0398) (Figure 6D). These results indicate that mAb14 interfered with the reduction in the IFN-γ secreted by pNK cells co-cultured with CTCL cell lines.

### 3.7. mAb14 Did Not Enhance the Cytotoxic Activity of pNK Cells against CTCL Cell Lines

The effect of mAb14 on the cytotoxic activity of pNK cells against CTCL cells was analyzed. CTCL (target) cells were stained with CFSE and then co-cultured with pNK (effector) cells at a ratio of 1:1 and 1:3 for 6 h. Viable and dead CTCL cells were analyzed using flow cytometry for 7AAD^+/−^CFSE^+^. An increase in the cell death of CTCL cells was observed when co-cultured with pNK cells compared to when cultured alone (Figure 7A), indicating the killing activity of pNK cells against CTCL cells. Unexpectedly, repeating the same assay but with the addition of mAb14 to the co-cultured cells did not increase the percentage of lytic CTCL cells (7AAD^+^CFSE^+^) compared to the isotype controls (Figure 7B–D).

## 4. Discussion

The overexpression of PCNA in the cell nucleus is known to correlate with cancer virulence (necessary for tumor survival), DNA replication, DNA repair, chromatin structure maintenance, chromosome segregation, and cell cycle progression [54,55]. In this study, it was observed that PCNA is expressed not only in the cell nucleus, but also on the cell membrane and in the cytoplasm of CTCL cell lines and PBMCs from patients with SS. The levels of csPCNA were found to vary in different CTCL cell lines, with a high expression in SS cell lines and a low expression in skin-derived MF and aggressive CTCL cell lines. This disparity could potentially be attributed to the distinct nature of SS and MF, as was recently evidenced by Harro et al., who demonstrated a distinct phenotype and differentiation trajectory for MF and SS, evidenced by their TCR repertoires and mutation profiles [67]. This difference may be attributable to differences in the nuclear-to-cytoplasmic relocalization of PCNA, as reported in neutrophils [68], or the variable expression of Annexin A2, a membrane that traffics protein to lipid rafts [61,68], as reported in other cancers [60,61]. Previous studies of CTCL have demonstrated the co-localization of PCNA with TOX to cytoplasmic and membrane regions [62]. Moreover, we found that the expression of csPCNA in CTCL was not correlated with apoptosis or cell death, but it was associated with the G2/M phase and cell cycle progression.

nPCNA is rarely expressed in stationary cells. It is synthesized in the G1 phase, peaks in the S phase, and significantly decreases in the G2/M phase [69]. Therefore, we assume that nPCNA accumulates during DNA synthesis in the S phase and then translocate to the cell membrane during G2/M.

csPCNA is a potential malignant marker for SS due to our three major findings: (1) The expression of csPCNA was observed in the malignant enriched cell fraction of SS PBMCs (viable CD4+CD26-) [42,70,71], which are characteristic of malignant cells. (2) Normal counterpart cells of healthy controls (viable CD4^+^CD26^−^) and non-malignant SS PBMCs (non-CD4+CD26-) did not show csPCNA expression. (3) Considering the atypical cerebriform morphology of the CD4^+^CD26^−^ SS PBMCs expressing csPCNA [37,66,70], advanced studies are needed to evaluate the correlation between csPCNA expression and mAb14 in the biopsies and blood samples of patients with MF, and the treatment response and clinical parameters.

Activated NK cells are primary rapid and potent producers of IFN-γ during the early innate immune response, which shapes the type and quality of the adaptive immune response that is subsequently elicited in cancer [14]. The secretion of IFN-γ was significantly lower in CTCL cell lines than in pNK cells. This finding is consistent with previous reports showing the poor secretion of IFN-γ by CTCL lines [72,73]. In this study, CTCL cell lines showed a poor secretion of IFN-γ compared to pNK cells. Additionally, CTCL cell lines inhibited the secretion of IFN-γ by pNK cells (mostly by the SS cell line), as supported by other studies showing TLR signaling genes in NK cells from patients with SS down-regulated type I (IFN-α/β) and type II (IFN-γ) interferon [45,74]. However, treatment with mAb14 increased IFN-γ secretion in pNK cells when co-cultured with CTCL cell lines, in accordance with reports on various leukemic and solid tumor cells [53]. Despite the increased secretion of IFN-γ, this study found that pNK cells did not exert a significant cytotoxic effect on CTCL cells in the presence of mAb14. This may be due to the presence of inhibitory ligands that counteract the activating signals on NK cells or the involvement of other regulatory factors in NK cell cytotoxicity. Kundu et al. [53] demonstrated that mAb14 enhanced the NK-mediated lysis of solid and leukemic cancer cell lines and tumor cells in patient-derived xenografts in mice. Herein, the lack of mAb14-mediated pNK cytotoxicity against CTCL cells might be due to the presence of other inhibitory ligands that correspond to inhibitory NK receptors [75], or other co-ligands that are required for NK cytotoxic activity, such as HLA I [76]. Given that NKp44 is only one potential regulator of NK cells, other proteins might play a role in regulating the killing activity of NK cells.

The findings suggest that csPCNA could be a relevant marker for malignancy in SS and may have implications for diagnosis and treatment, with a potential similar role in other CTCL variants. Additionally, this study highlights the complex interplay between CTCL cells and the immune response, particularly the role of IFN-γ and NK cells, which could be targeted for therapeutic interventions.

## 5. Conclusions

We have identified csPCNA as a potential specific diagnostic biomarker for CTCL, with a particular emphasis on SS. We encourage further studies on the use of mAb14 as a diagnostic tool for peripheral blood involvement in SS and other CTCLs, skin biopsies, therapeutic monitoring, residual disease, disease progression, and prognosis. Additionally, an in vivo mouse model of CTCL with an activated immune arm of NK cells may elucidate the potential of mAb14 for NK-mediated immunotherapy in CTCL and its application in antibody–drug conjugate therapy.

## Figures and Tables

**Figure 1 cancers-15-04421-f001:**
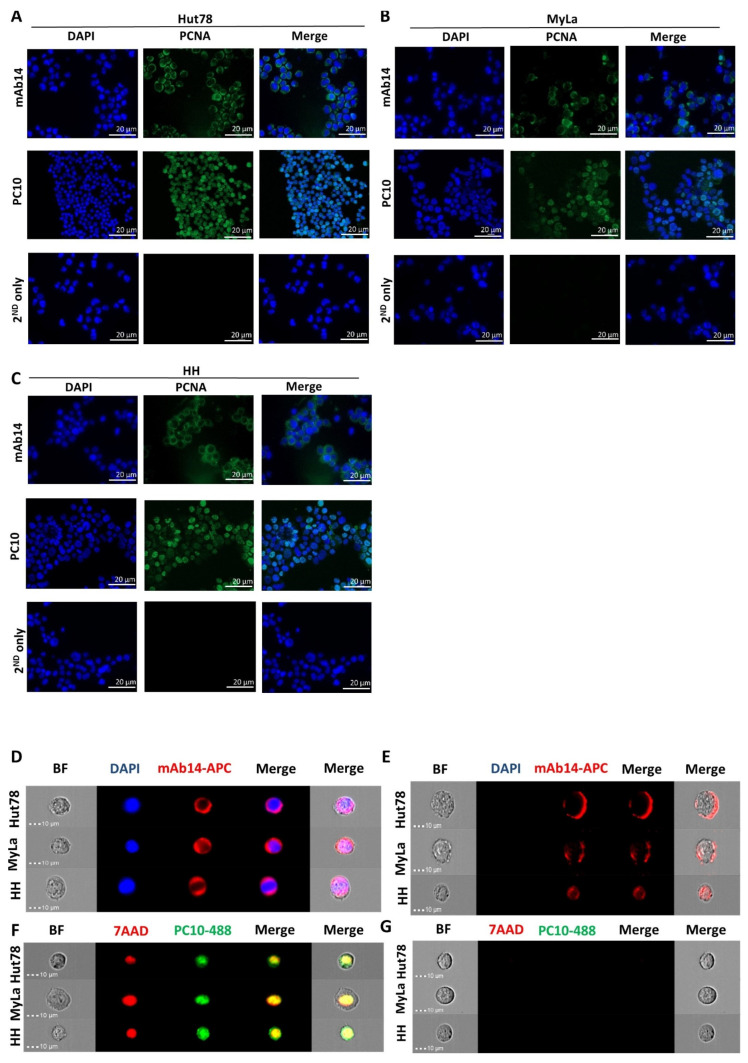
mAb14 antibody detects PCNA on the membrane and in the cytoplasm of CTCL cell lines, while PC10 detects PCNA in the nucleus of the cells. Hut78 (**A**), MyLa (**B**), and HH (**C**) cells were stained for csPCNA with mAb14 antibody (**upper panel**), nPCNA was stained with PC10 antibody (**middle panel**), and secondary antibody SA-APC was used as a control (**lower panel**). All cells were co-stained with DAPI, and images were obtained using the Apotome microscope ×40. Fixed and permeabilized cells were stained with mAb14 (red) (**D**), and PC10 was (green) (**E**) co-stained either with DAPI (blue) or 7AAD (red) and analyzed using the flow cytometer ImageStream ×40. A similar analysis of (**D**) and (**E**) was performed in un-fixed un-permeabilized cells stained with mAb14 (**F**) and PC10 (**G**).

**Figure 2 cancers-15-04421-f002:**
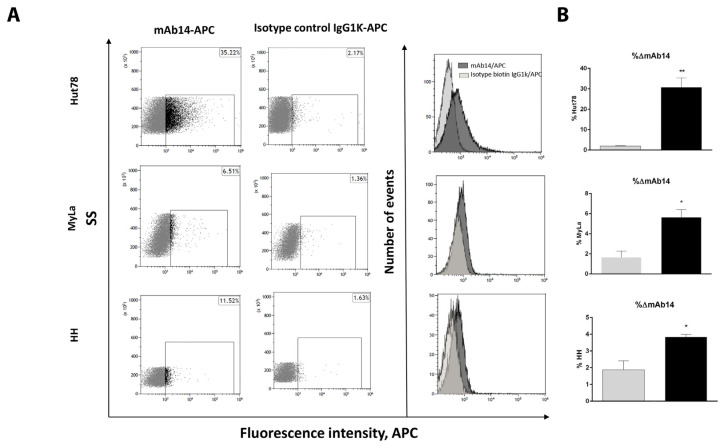
More than 80% of CTCL cells in the CTCL cell lines expressing csPCNA, as detected by mAb14, were viable and non-apoptotic cells. CTCL cell lines were analyzed using a flow cytometer for staining with mAb14 vs. the isotype IgG1k on un-fixed cells without permeabilization (**A**); column curve for %mAb14^+^ vs. the isotype control in each cell line (**B**), (n = 3). Cell lines were stained with mAb14 (**C**,**F**) or the isotype control IgG1ĸ (**D**,**F**), together with annexin-FITC and PI (**E**,**F**). Positive cells for mAb14 are shown in black within the scatter plot. The statistical significance was determined between the percent of positive cells for mAb14 and the isotype IgG1k, and between the viable vs. unviable mAb14+ cells in each cell line based on the paired *t*-test (* *p* < 0.05; ** *p* < 0.01; *** *p* < 0.001). Error bars for mean ± SD.

**Figure 3 cancers-15-04421-f003:**
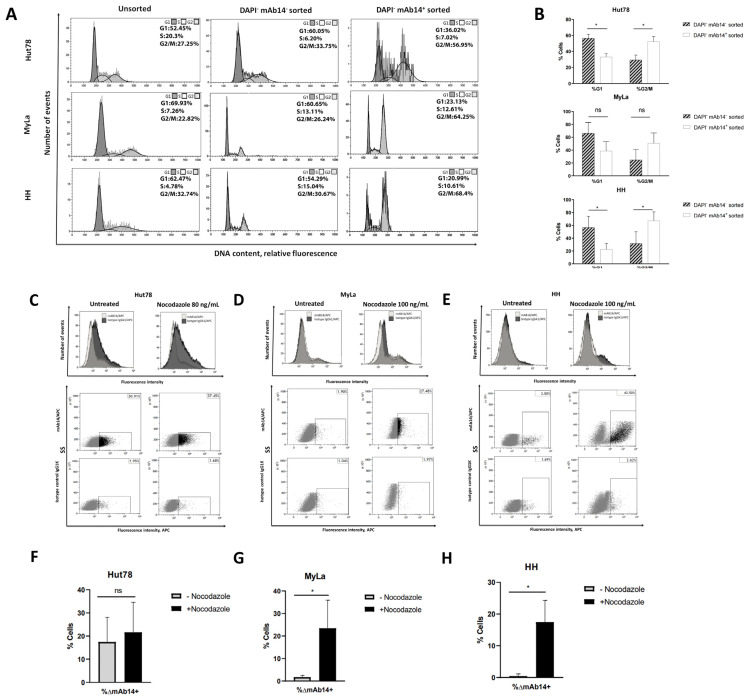
Expression of csPCNA in CTCL cell lines, as detected by mAb14, is associated with the G2/M phase. CTCL cell lines were sorted using FACS for mAb14^+^DAPI^−^ cells and for mAb14^−^DAPI^−^ cells. The unsorted (**left panel**) and sorted cells, with DAPI^−^ mAb14^−^ (**middle panel**) and DAPI^−^ mAb14^+^ (**right panel**), were subjected to cell cycle analysis (**A**). The percent of mAb14^+^DAPI^−^ vs. DAPI^−^ mAb14^−^ cells in the G1 phase and G2/M phase is shown, n = 3 (**B**). A Some of the untreated and Nocodazole-treated cells were stained with mAb14 and the isotype control; this is presented in the mean fluorescence intensity and the percent of positive cells (**C**–**E**). The percent of mAb14^+^ in the untreated and Nocodazole-treated cells is shown in the column curves, n = 3 (**F**–**H**). The statistical significance was determined as the percent of ∆mAb14+ in the un-treated cells vs Nocodazole-treated cells, based on the un-paired *t*-test (ns = no significant increase; * *p* < 0.05 and error bars for mean ± SD.

**Figure 4 cancers-15-04421-f004:**
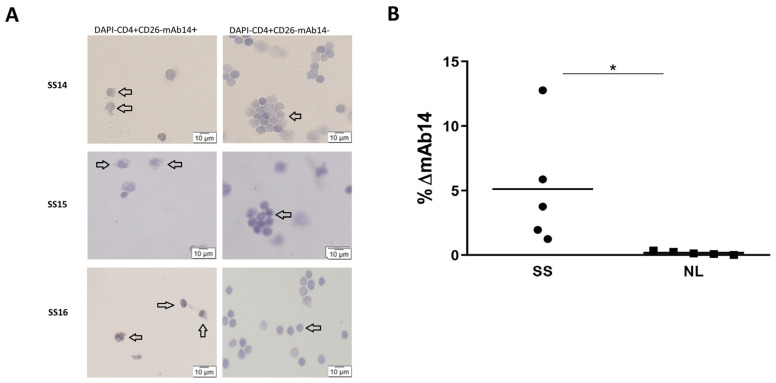
mAb14+DAPI-CD4+CD26- cells of SS PBMCs have an atypical morphology compared to the normal morphology of their counterpart mAb14-cells. PBMCs were isolated from blood samples of SS patients, and sorted using the FACS sorter for mAb14+DAPI-CD4+CD26^−^ cells and for mAb14-DAPI-CD4+CD26-cells. The sorted cells were cytospined on slides and then stained with H&E. The representative light microscope images X40 are presented, n = 3 (**A**). A percent of the DAPI^−^CD4^+^CD26^−^ positive for the mAb14 minus isotype control in SS patient’s vs. healthy donors is presented in the scatter plot, n = 5 (**B**). The statistical significance was determined as the DAPI+CD4+CD26^−^ positive for mAb14+ related to the isotype control for five SS PBMCs vs. PBMCs from five healthy donors, with significance based on the un-paired *t*-test (* *p* < 0.05).

**Figure 5 cancers-15-04421-f005:**
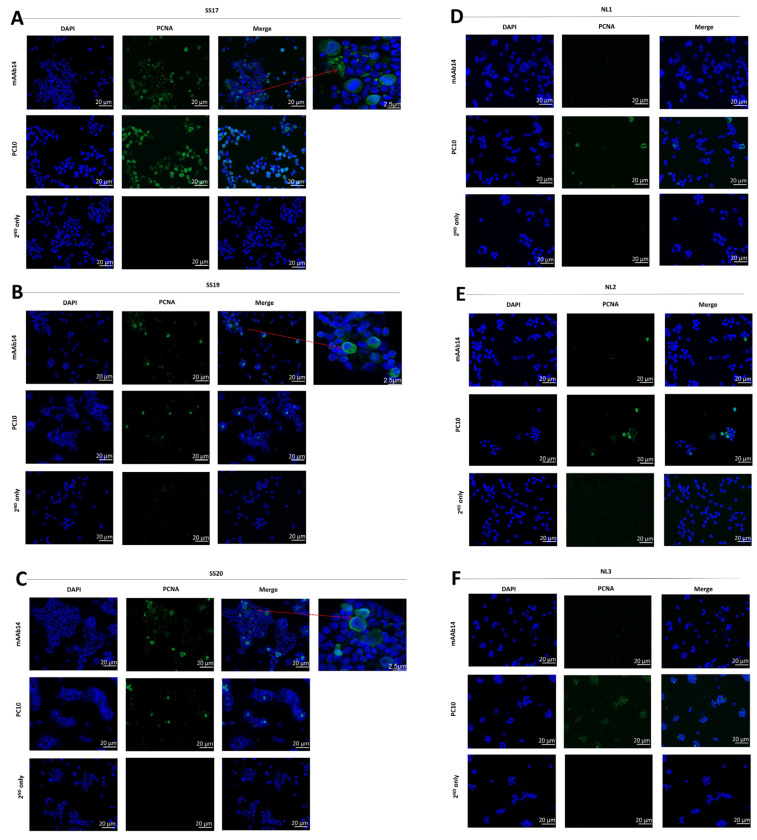
mAb14 detects PCNA in the cytoplasm and membrane of Sezary PBMCs specifically on CD4^+^CD26^−^ cells, with no detection in healthy controls. Fixed and un-permeabilized PBMCs isolated from three SS patients (**A**–**C**) and three healthy controls (**D**–**F**) were stained for csPCNA with the mAb14 antibody (**upper panel**), for nPCNA with the PC10 antibody (**middle panel**), and for only the secondary antibody (**lower panel**). All cells were co-stained with DAPI, and images were taken using the Apotome microscope ×40 with a size bar of 20 µm and a zoom in of positive staining with a size bar of 2.5 µm (**A**–**C**). Five SS PBMCs (**G**) and five healthy donors (**H**) were stained with CD4-FITC, CD26-PE, DAPI and mAb14 or isotype mIgG1ĸ with APC. The percent of DAPI^−^CD4^+^CD26^−^ on lymphogated cells, considered as SS-enriched cells, is presented in the left panel (**G**,**H**). The percent of mAb14^+^ cells gated on DAPI^−^CD4^+^CD26^−^ lymphocytes and the percent of mIgG1^+^ cells gated on DAPI^−^CD4^+^CD26^−^ lymphocytes are presented in the middle panels (**G**,**H**). csPCNA (detected by mAb14) is expressed on a fraction of the CD4^+^CD26^−^ peripheral lymphocytes of SS patients, but not on CD4^+^CD26^−^ lymphocytes from healthy individuals.

**Figure 6 cancers-15-04421-f006:**
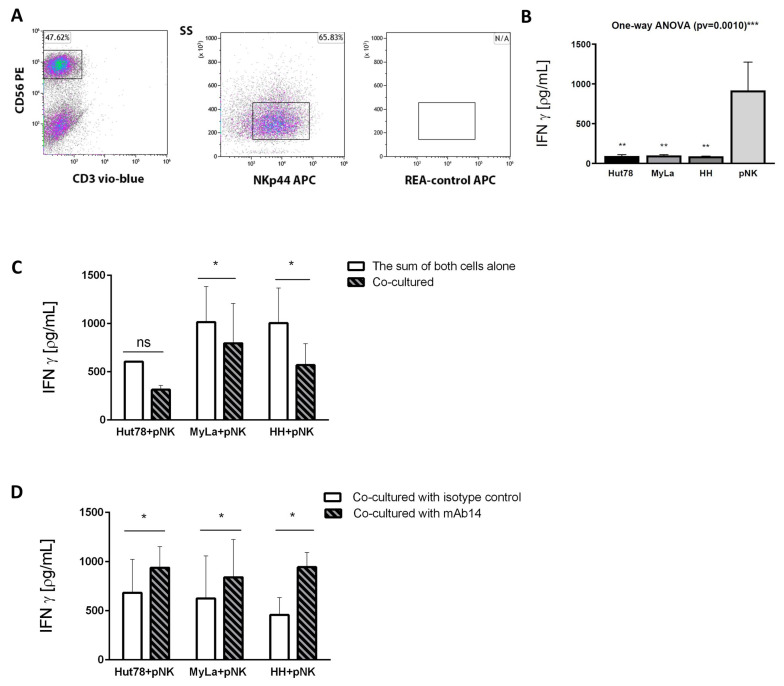
CTCL cell lines suppress the secretion of IFN-γ by pNK cells, and mAb14 reverses this effect. pNK cells were isolated from healthy donors, stained with CD3-Vio-blue, CD56-PE and NKp44-APC, and analyzed using flow cytometry (**A**). Distribution of CD3 and CD56 among pNK population (left panel), levels of NKp44 expression gated from CD3^−^/CD56^+^ population (middle panel), and REA control gated from CD3^−^CD56^+^ population (right panel). The IFN-γ secreted by pNK cells and CTCL cell lines that were cultured separately is presented (**B**). One-way ANOVA was performed among the groups; a comparison was performed between each pNK and each CTCL cell line (ns = no significant increase; * *p* < 0.05; ** *p* < 0.01. pNK cells from healthy donors and CTCL cell lines were cultured separately and in co-culture. IFN-γ secretion was assayed using ELISA in cell supernatant, and the sum of the IFN-γ level from the two separated cell cultures was compared to the level of IFN-γ in their co-culture (n = 3) (**C**). The level of IFN-γ was analyzed using ELISA in the pNK and CTCL cells co-cultured and treated with mAb14 compared to the isotype control (n = 4) (**D**). The statistical significance was determined as the sum of the IFN-γ values secreted by each cell alone compared to their co-culture, based on the paired *t*-test (ns = no significant increase; * *p* < 0.05), based on one way ANOVA (*** *p* < 0.001) and error bars for mean ± SD.

**Figure 7 cancers-15-04421-f007:**
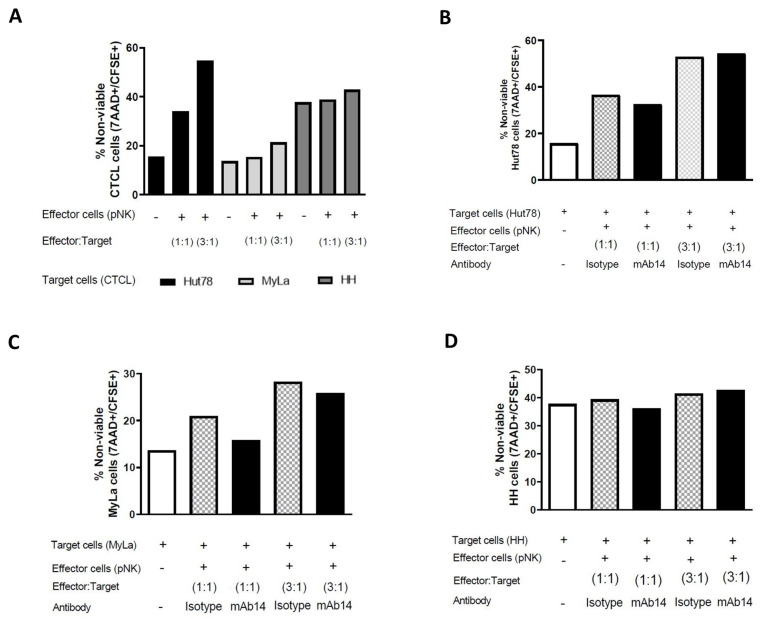
Co-culturing CTCL cells with pNK cells resulted in an increase in cell death, while mAb14 showed no effect on their cell viability. CTCL cell lines stained with CFSE were co-cultured with pNK cells at a ratio of 1:1 and 1:3 for 6 h, stained with 7AAD and analyzed using a flow cytometer for dead cells (CFSE^+^7AAD^+^) (**A**). The assay was performed once (n = 1). The experiment was repeated with the addition of mAb14 or the isotype control in Hut78 cells (**B**), MyLa cells (**C**), and HH cells (**D**). The statistical significance was determined as the percent of positive cells for mAb14 and the isotype IgG1k from all cell lines co-cultured at different ratios, with pNK cells, (n = 1).

## Data Availability

The data that support the findings of this study are available from the corresponding author upon reasonable request.

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
