# Peer review of "mAb14, a Monoclonal Antibody against Cell Surface PCNA: A Potential Tool for Sezary Syndrome Diagnosis and Targeted Immunotherapy"

_cancers, 2023, doi:10.3390/cancers15174421_

Round 1

Reviewer 1 Report

This is a sound, professionally researched paper on the biology and the role of mAb14 in cutaneous T-cell lymphoma. However, I have a few suggestions to improve this paper. 

1. Mostly, cutaneous T cell lymphomas, especially Mycosis fungoides, are diagnosed based on clinical and pathologic (skin biopsy) findings. 

It would be nice to demonstrate how mAb14 is expressed in FFPE skin tissue sections because it is the conventional way of an actual diagnosis. This way, the authors could strengthen their conclusion - mAb14 aid in diagnosing PCTL.

2. One minor point, on page 8, lines 279-280. 

 How did you recognize them as atypical lymphoid cells? I suggest adding a description of atypical cell morphology.

Author Response

Thank you for the insightful review of our manuscript. We are hopeful that our edited manuscript version meets the criteria for publication in the Cancers journal.

  1. It would be nice to demonstrate how mAb14 is expressed in FFPE skin tissue sections because it is the conventional way of an actual diagnosis. This way, the authors could strengthen their conclusion - mAb14 aid in diagnosing PCTL.

We agree that immunostaining of mAb14 on FFPE MF skin biopsies is crucial to demonstrate its diagnostic potential. However, we have refrained from doing it initially because mAb14 stains nuclear PCNA in addition to membrane and cytoplasmic PCNA on FFPE sections. Consequently, we have exclusively employed live cell staining methods, such as cytospinned cells and FACS, to specifically detect membrane PCNA.

  1. One minor point, on page 8, lines 279-280. How did you recognize them as atypical lymphoid cells? I suggest adding a description of atypical cell morphology.

A description was added and a relevant reference as follow:

The morphology of the DAPI-CD4+CD26- mAb14+ cells were mostly convoluted and atypical with moderately to highly grooved (ie, cerebriform) nuclei and high nuclear to cytoplasmic ratio as typical for the lymphoma cells of SS.68 While the DAPI-CD4+CD26- mAb14- subpopulation was mostly a mixture of round shape non-cerebriformic normal lymphocytes.

Reference 68 was added to support the recognition of SS cells: Vonderheid EC. On the diagnosis of erythrodermic cutaneous T-cell lymphoma. J Cutan Pathol. 2006 Feb;33. Suppl 1:27-42.

Thank you for considering our study for publication in Cancers journal

Lilach

Reviewer 2 Report

The article by Knaneh and colleagues propose the use of a monoclaonal antibody capable to recognize PCNA, on the membrane of unfixed, unpermeabilized cell derived from T-cell lymphoma as a potencial marker of the disease. The authors indicate that could be use as a marker since they detect this protein in malignant cells derived from patients with T-cell lymphoma and not in healthy donors. The reduce n of human tested makes it difficult to state this with confidence.

There are several issues in the presentation and the quality of the data that do not convince this reviewer that this article is ready to be considered for publication.

In general, the figures have such a low resolution that not even with a 250% increase in the size of the page is possible to read the labels, the axis label or to clearly distinguished the staining’s. This is particularly frustrating on the flow cytometry graphs.

Except for figure 1D/E none of the images have a bar size, and even in some of the panels for the same cell line the nucleuses have different sizes, which suggest that maybe not all images were acquire with the same settings. The color chosen are very difficult to see, maybe the authors can change the pseudocolor, or present the single channels as black and white and inverted images.

In none of the figure legends the authors indicate what * represent, not in all of them the n of the experiment is shown. For example, in figure 2H is not possible to read the axis and is not clear what is been compared to what, the bars 1 and 2 are at the same level, so what are the **? Or are they ***?

In figure 3 some the graph cannot be properly read, and some have *, others indicate ns, while others nothing. Figure 6 does not have error bars; shall we assume that is n=1? These are is just two example of the lack of care in the presentation of this paper.  

In figure 5, how is it OK to add the sum of the interferon from each cell on the graph, and not showing them separate, specially since they describe each phenotype separately, is very confusing.

There are two sections of the results referring to supplementary figures, if each one of those figures deserve a subsection why are they not in the amin text? Also Fig S1 has an *, but the error bar is the same size than the actual bar, basically 5 +/-5, how can that be significant, if that * means significance?

The introduction is quite short, does not properly explain the normal function of PCNA, neither how (if known) is possible that goes to the membrane.

Not until the discussion the authors indicate what kind of CTCL each line represents, this should be described in the methods.

The results section is quite confusing, especially when the author try to explain every time the subtraction of the isotype on the numbers. I would suggest to be more straight forward, when explaining the results.

Another issue is that the figure legends, described the experiment, not the result they are trying to represent.

In summary, the low quality of the figures, the low consistency on the presentation of the results, the dubious statistical analysis and the complex explanation of the results suggest to me that this article needs a lot of improvement before to be considered suitable for publication.

Author Response

Reviewer 2

Thank you for the insightful review of our manuscript, and We are hopeful that our edited manuscript version meets the criteria for publication in the Cancers journal. Please find our reply point by point.

  • In general, the figures have such a low resolution

We apologize for the inconvenience. It occurred due to a mistake when we attempted to lower the file size. All pictures have now been replaced with high-resolution images at 300 DPI.    

  • Except for figure 1D/E none of the images have a bar size

Bar size were added to all the microscopic images (Fig: 1,4,5). 

  • In some of the panels for the same cell line the nucleuses have different sizes, which suggest that maybe not all images were acquire with the same settings

We revised our images and they were all taken in the same setting.

  • The color chosen are very difficult to see, maybe the authors can change the pseudocolor, or present the single channels as black and white and inverted images

We changed the pseudo color from red to green. 

  • In none of the figure legends the authors indicate what * represent

We have added in the figure legends explanation for all symbols of significance and specified the statistical tests used for the relevant compared groups.

  • Not in all of them the n of the experiment is shown.

The number of repeated experiments were added in all figure legends.

  • For example, in figure 2H is not possible to read the axis and is not clear what is been compared to what, the bars 1 and 2 are at the same level, so what are the **? Or are they ***? 

Fig 2H has been replaced with a new figure that includes clear axis legends. Additionally, a matched column of non-viable cells has been added to the comparison with the viable cells. Furthermore, the figure legends now provide a detailed explanation of the significance.

  • In figure 3 some the graph cannot be properly read, and some have *, others indicate ns, while others nothing.

Graphs and scatter plots of figure 3 have been replaced with new clear ones. Additionally, a missing significance symbols were added with detailed explanation in the figure legends. 

  • Figure 6 does not have error bars; shall we assume that is n=1 

We present here result of one experiment, and we added that information in the revised figure legends. (Fig 7 in the revised manuscript).

  • In figure 5, how is it OK to add the sum of the interferon from each cell on the graph, and not showing them separate, specially since they describe each phenotype separately, is very confusing.

An explanation was added to the text as follow:

Since, the IFN-γ level in the co-culture of pNK cells with CTCL cell line is attributed by the two cells with no option to discriminate between them, we compared it to the sum of IFN-γ secretion by each cell alone.

(Fig 6 in the revised manuscript)

  • There are two sections of the results referring to supplementary figures, if each one of those figures deserve a subsection why are they not in the amin text?

We have included the supplementary figures in the main text as figure 4A and 4B.

  • Also Fig S1 has an *, but the error bar is the same size than the actual bar, basically 5 +/-5, how can that be significant, if that * means significance?

We replaced the column graph with a scatter plot, as it better represents the data than error bars. The significance was p<0.05 added in the graph and in figure legends. (Fig 4 in the revised manuscript)

  • The introduction is quite short, does not properly explain the normal function of PCNA, neither how (if known) is possible that goes to the membrane.

We have added a whole new paragraph on markers for CTCL, including their 36 citations. Furthermore, we have provided a possible explanation of how PCNA migrates to the membrane, along with its application in suppressing NK cells, and supportive data from other cancers.

  • Not until the discussion the authors indicate what kind of CTCL each line represents, this should be described in the methods.

Description of the 3 CTCL cell lines was added in the material and methods chapter. 

  • The results section is quite confusing, especially when the author try to explain every time the subtraction of the isotype on the numbers. I would suggest to be more straight forward, when explaining the results.

The following mathematical notation was added in the materials and methods: ∆ positive staining for mAb14 = (%mAb14+ cells) – (%isotype control+ cells), and it refers only to the values of ∆ mAb14. All other values of %mAb14 are absolute values that were statistically compared to the isotype control, but they were not calculated as deltas.   

  • Another issue is that the figure legends, described the experiment, not the result they are trying to represent.

The titles of all the figures were replaced from experiment description to result description.

Thanks you for considering our study.

Lilach

Reviewer 3 Report

Although authors have done good job by studying the expression of csPCNA detected by mAb14 in  primary 19 cutaneous T-cell lymphoma. However, the following comments will make the manuscript comprehensive.

1. The figures particularly first four figures needs better resolution, better legends to read and interpret properly and then only more comments can be suggested.

2. The English and scientific language needs to be improved.

3. The patient collection information, consent, ethics and methods part needs to be elaborate.

4. The surface markers for CTCL like CD30 or Inducible T-cell costimulator (ICOS) should be used as positive control or as co-expression for csPCNA.

5. More relevant and latest references needed to be added.

The English and scientific language needs to be improved.

Author Response

Reviewer 3

Thank you for the insightful review of our manuscript. We are hopeful that our edited manuscript version meets the criteria for publication in the Cancers journal. Please find our point by point reply. 

  1. The figures particularly first four figures needs better resolution, better legends to read and interpret properly and then only more comments can be suggested.

We emphasize our apology for the inconvenience. This was occurred due to a mistake when we attempted to lower the file size. All pictures have now been replaced with high-resolution images at 300 DPI.

  1. The English and scientific language needs to be improved.

English improvement was done and some sentences and paragraphs were rephrased and all are highlighted in gray.

  1. The patient collection information, consent, ethics and methods part needs to be elaborate. 

We added the following information: A table of patient clinical data was added in the supplementary; Ethics approval numbers; and methods for isolation of PBMCs.

  1. The surface markers for CTCL like CD30 or Inducible T-cell costimulator (ICOS) should be used as positive control or as co-expression for csPCNA.

Indeed many markers have been suggested to identify the malignant blood lymphocytes, and a whole paragraph discussing it was added in the introduction. However, the ISCL/USCLC/EORTC consortium currently recommends using only CD4+CD7- and/or CD4+CD26- lymphocytes to determine blood involvement. This was added in the introduction along with the relevant reference #42: Elise A Olsen, Sean Whittaker, Rein Willemze et al. Primary cutaneous lymphoma: recommendations for clinical trial design and staging update from the ISCL, USCLC, and EORTC. Blood, 2022 Aug 4;140(5):419-437.

  1. More relevant and latest references needed to be added.

As part of the reviewer's request for expanding the introduction and making several clarifications, 54 references were added. Among these references, 4 are  latest ones (#32,34,39,53).

Thank you for considering our study

 Lilach

Round 2

Reviewer 1 Report

The authors have done well revising the manuscript.

Minor comment

English editing is required including - 

Page 2 line 49 various molecular biomarkers associated with CTCL.

Author Response

Dear Reviewer

We greatly appreciate the thoughtful assessment of our manuscript. In response to the reviewers' comments, we have made revisions to the manuscript and addressed all inquiries as follows:

  1. English editing is required including,
  2. Page 2 line 49 various molecular biomarkers associated with CTCL.

We applied English editing throughout the entire introduction including the specific sentence that was left incomplete. The modifications have been marked with underlines within the text.

Hopefully, our refined manuscript aligns with the requirements for publication in the Cancers journal.

With appreciation

Lilach Moyal

Reviewer 2 Report

This article has significantly improved its quality of presentation, but still the text is hard to follow in some sections and the message the authors are trying to deliver is not clear. Still some improving in figures quality can be done. I believe this article is not yet ready as it is to be accepted for publication. I have listed some specific comments, but I strongly suggest a thorough revision of the presentation of both the text and the figures.

Introduction

In general, the paragraphs included in this version and that are shown in yellow, are poorly written and somehow disconnected of the introduction flow.

Line 49. This paragraph seems to be missing its initiation. Then, there are capital letters after some , . But more importantly the paragraph seems to be disconnected and out of context, What is the message the authors want to give us by its inclusion in this new version of the paper?

Line 84. The second part of this paragraph is not well written, please check the grammar. It is very strange the manner the authors start this second part.

Material and methods

Ref. 7175  and Ref. 6515 are the bioethical permissions? If so, Why not state that?

Results

Figure 1F PC- in indicated, this should be PC-488

Figure 2 now takes 4 pages, authors could divide it on the two measeegs they rea giving here, one figure to show the % of cells were csPCNA can be detected, and a second figure to show that those cells are not apoptotic. Also Figures 1C and D seems redundant whit the results presented on figure 1.

Figure 3  shows clear results about the higher expression of csPCNA in G2-M, but the fact that has so many panels is distracting, to show that cells are indeed synchronized after nocodazole treatment  is a necessary control, but could be presented as a supplementary figure. The point the authors are making here is clear without having to show it in the main figure.

Figure 4 A, the cell staining has size bars that are draw as rectangles, and some are too small that cannot be read. Still the cells that present the specific characteristics mentioned in the text should be indicated (arrow heads for example). I am glad the authors are showing the actual distribution of the data, that is clear unlike the previous presentation.

In figure 5 A-F the authors are stating they show cytoplasmic and nuclear staining when using mAB14 (in the figure the label is mAAb14), and only nuclear when using mPC10 on cells derived from SS individuals. This is not very clear due to the objective use. If the authors still have the samples, can they take representative pictures with a higher objective, maybe 40 or 60X?  Rigth now it all seems nuclear. Also as shown the size bar is almost invisible, we can only see the 20um word, rather than the line. If a clear line is used, then you can state in the legend that size bar is 20um.

Lines 404-406 describe the setup but not the result, the following part of that paragraph is referred to INF secretion to pNK levels.

Author Response

Dear reviewer

We greatly appreciate the thoughtful assessment of our manuscript. We have made revisions to the manuscript and addressed all inquiries. The modifications have been marked with underlines within the text. Our reply is as follows:

In general, the paragraphs included in this version and that are shown in yellow, are poorly written and somehow disconnected of the introduction flow.

We aligned with the reviewer's input and subsequently rephrased the entire introduction chapter.

Line 49. This paragraph seems to be missing its initiation. Then, there are capital letters after some , . But more importantly the paragraph seems to be disconnected and out of context, What is the message the authors want to give us by its inclusion in this new version of the paper?

Apologies for the incomplete sentence. We edited its English and complete it as follow: “Numerous molecular biomarkers are associated with CTCL, providing significant insights into disease origin, accurate diagnosis, prognosis evaluation, and optimal treatment selection”.

Line 84. The second part of this paragraph is not well written, please check the grammar. It is very strange the manner the authors start this second part.
The entire paragraph underwent English editing and has been rephrased.  

 Ref. 7175  and Ref. 6515 are the bioethical permissions? If so, Why not state that?

We added a statement as follow:PBMCs were collected under local Helsinki approval for bioethical permission of Rabin Medical Center: Ref. 7175 for blood of 9 patients with SS and Ref. 6515 for leftover blood of healthy donors”.

Figure 1F PC- in indicated, this should be PC-48

We corrected it to PC-488

 Figure 2 now takes 4 pages, authors could divide it on the two measeegs they rea giving here, one figure to show the % of cells were csPCNA can be detected, and a second figure to show that those cells are not apoptotic. Also Figures 1C and D seems redundant whit the results presented on figure 1.

 Figure 2C and 2D were transferred to the supplementary section.

 Figure 3 shows clear results about the higher expression of csPCNA in G2-M, but the fact that has so many panels is distracting, to show that cells are indeed synchronized after nocodazole treatment  is a necessary control, but could be presented as a supplementary figure. The point the authors are making here is clear without having to show it in the main figure.

The figures of cell cycle arrest by Nocodazole (upper panel of fig 3C,D,E and fig 3F,G,H) heve been relocated to the supplementary section. The figure is indeed clearer without it.

 Figure 4 A, the cell staining has size bars that are draw as rectangles, and some are too small that cannot be read. Still the cells that present the specific characteristics mentioned in the text should be indicated (arrow heads for example). I am glad the authors are showing the actual distribution of the data, that is clear unlike the previous presentation.

The text of size bars in fig 4A was increased. Arrow heads for specific aberrant cells and normal cells were added.

 In figure 5 A-F the authors are stating they show cytoplasmic and nuclear staining when using mAB14 (in the figure the label is mAAb14), and only nuclear when using mPC10 on cells derived from SS individuals. This is not very clear due to the objective use. If the authors still have the samples, can they take representative pictures with a higher objective, maybe 40 or 60X?  Rigth now it all seems nuclear. Also as shown the size bar is almost invisible, we can only see the 20um word, rather than the line. If a clear line is used, then you can state in the legend that size bar is 20um.

We have incorporated zoom-in images of specific cells exhibiting cytoplasmic/membrane staining. We would like to express our gratitude to the reviewer for making this crucial suggestion, which has significantly enhanced both our manuscript and our main findings.

We have also emphasized the line bars in figure 5 as well as in figure1.  

Lines 404-406 describe the setup but not the result, the following part of that paragraph is referred to INF secretion to pNK levels.
The setup description has been moved to the materials and methods section, while the results text now exclusively emphasizes the results data.

Thank you for considering our study

With appreciation

Lilach Moyal

Round 3

Reviewer 2 Report

This version of the manuscript is clear, the message the authors want to deliver is clear, and the data is clearly presented. Thank you for accepting my suggestions.